# Time Trends in Psoriasis and Psoriatic Arthritis Incidence from 2002 to 2016 in Taiwan: An Age–Period–Cohort Analysis

**DOI:** 10.3390/jcm11133744

**Published:** 2022-06-28

**Authors:** Yu-Tsung Chen, Chih-Yi Wu, Yu-Ling Li, Li-Ying Chen, Hung-Yi Chiou

**Affiliations:** 1School of Public Health, College of Public Health, Taipei Medical University, Taipei 110, Taiwan; sunshinef91@yahoo.com.tw; 2Department of Dermatology, Shuang Ho Hospital, Taipei Medical University, New Taipei City 235, Taiwan; 3Institute of Population Health Sciences, National Health Research Institutes, Miaoli 350, Taiwan; jmps7129@gmail.com (C.-Y.W.); evelyn.tmu@gmail.com (Y.-L.L.); 4Health Data Analytics and Statistics Center, Office of Data Science, Taipei Medical University, Taipei 110, Taiwan; lychen@tmu.edu.tw; 5Master Program in Applied Epidemiology, College of Public Health, Taipei Medical University, Taipei 110, Taiwan

**Keywords:** psoriasis, psoriatic arthritis, incidence, epidemiology, age–period–cohort analysis

## Abstract

Background: Psoriatic disease is a chronic inflammatory disease that is associated with morbidity and a poor quality of life. However, studies on the trends of psoriatic disease incidence are limited. We examined trends in psoriasis and psoriatic arthritis from 2002 to 2016 in Taiwan and distinguished the effects of age, period, and cohort on those trends. Methods: Data from the National Health Insurance Research Database were analyzed for the annual incidence of psoriasis and psoriatic arthritis. An age–period–cohort model was designed in order to investigate the effects of each age, period, and birth cohort on the incidence. Results: From 2002 to 2016, the incidence of psoriasis significantly decreased from 43.33 to 23.14 per 100,000 persons. The incidence of psoriatic arthritis significantly increased from 3.57 to 5.22 per 100,000 persons. In the age–period–cohort analysis, the net age effect on the incidence of psoriasis and psoriatic arthritis increased with advancing age (6-fold and 7.7-fold difference, respectively). Conclusion: The age–period–cohort analysis revealed that the incidence of psoriasis and psoriatic arthritis is associated with older age and early birth cohorts. Elderly individuals in Taiwan may be at a higher risk of developing new-onset psoriasis and psoriatic arthritis.

## 1. Introduction

Psoriasis (PsO) is a chronic inflammatory skin disease [1] that is associated with morbidity and a poor quality of life. Approximately one-third of PsO cases develop psoriatic arthritis (PsA), which is a seronegative arthritis that is characterized by arthritis, enthesitis, dactylitis, and nail abnormality. The prevalence of PsO varies by ethnicity and geographic location [2,3]. It is more common in higher latitudes than in tropical regions [4] and is less prevalent in Asians [5,6]. PsO develops through multiple risk factors, including genetic predisposition and interaction with environmental and immunological factors that are specific to ethno-racial groups [7,8]. Notably, the expression of the most strongly associated susceptibility gene, human leukocyte antigen (HLA)-Cw6 locus, is lower in Taiwanese versus Caucasian patients [9].

Relatively, studies on the incidence of PsO and PsA are limited, particularly in Asian regions. Most of the studies that have been reported thus far have been performed in Western countries [10,11,12]. The previous studies have primarily used unadjusted age for analysis and have rarely examined the effects of age, period, and cohort on the prevalence or incidence of PsO. It is currently unclear whether the increasing trend in the prevalence of PsO reflects the changes across calendar periods or birth cohorts, independent of population aging. Age–period–cohort (APC) analysis is an effective method for identifying the independent effects of age, period, or cohort on the trends of disease incidence and mortality. It estimates, in addition to the effects of the patient’s age, the individual effects of the time period, or the patient’s birth cohort, and the biological, socioeconomic, and health factors that affect a specific population [13,14,15], which are usually omitted in cross-sectional studies. The age effect represents the association between the risk of disease and aging. The period effect represents the external factors that simultaneously affect all of the age groups at a particular calendar time, such as the development of a screening program or diagnostic technique. The cohort effect represents a change that characterizes the populations born at the same period but is independent of the process of aging. For example, the risk of cardiovascular disease increases with age. People that are born in the same time period can have a similar lifestyle, which may increase the risk of cardiovascular disease; this is a cohort effect. Consequently, the APC model has been developed in order to estimate the independent effects of these three components. Understanding the trends of disease can help us to allocate medical care and improve healthcare policy. The aim of this study was to examine the trends in psoriatic disease from 2002 to 2016 in Taiwan and to distinguish the effects of age, period, and cohort on those trends.

## 2. Materials and Methods

### 2.1. Data Source

This study was approved by the Taipei Medical University Joint Institutional Review Board (No. 202002033). Data from the National Health Insurance Research Database (NHIRD), covering >99.9% of nearly 23 million people in Taiwan, were assessed. The NHIRD contains registration files and the original reimbursement claims data, including demographic characteristics, dates of admission and discharge, diagnostic codes, procedures performed, and the details of prescriptions and comorbidities [16]. This database has been widely used in epidemiological studies and has exhibited high accuracy and validity in recording PsO [17].

### 2.2. Study Population

From 1 January 2002 to 31 December 2016, patients with at least three consecutive outpatient visits, or one admission claim to any medical setting, for psoriatic disease were included. Patients with prior PsO diagnosis before 2002 were excluded. Psoriatic cases were defined as those with a diagnostic code for PsO or PsA (ICD-9-CM of 696.0 for psoriatic arthropathy or 696.1 for other PsO; ICD-10-CM of L40.0–L40.4, L40.8, or L40.9 for PsO or L40.50–L40.54, or L40.59 for arthropathic PsO) that were assigned by a dermatologist or rheumatologist. The date of first diagnosis of PsO was assigned as the index date. Patients who had received systemic therapy or phototherapy were classified as severe PsO and the remaining patients were classified as mild PsO.

### 2.3. Outcomes

Data were analyzed for the annual prevalence and incidence of PsO and PsA. Prevalent cases were those meeting the aforementioned criteria for each year. Incident cases were those that met the criteria with a primary diagnosis of PsO. Patients who lost their beneficiary status in the NHIRD were excluded from the analysis. Data were arranged into consecutive 5-year periods for the APC analysis. Additionally, patients aged 10–69 years were subdivided into 5-year age groups.

### 2.4. Statistical Analysis

The annual percentage change in PsO/PsA incidence and prevalence were evaluated using joinpoint regression analysis. The number of changepoints, termed joinpoints, was estimated using permutation tests [18]. We used the log-linear model under the Poisson assumption. This analysis was performed using “Joinpoint Regression Program 4.9.0.0” from the Surveillance Research Program of the US National Cancer Institute.

Age-standardized incidence rates per 100,000 persons (based on the World Health Organization 2000 Standard Population) were calculated for the study period to identify the long-term trends. A generalized linear model was used under the assumption that the response variable, the incidence of PsO or PsA, follows a Poisson distribution. The hierarchy of models and sequential statistical tests proposed by Clayton and Schifflers were used to evaluate age, period, and cohort effects [19,20]. We performed three one-factor models (age, period, and cohort models), three two-factor models (age–period, age–cohort, and period–cohort models), and the three-factor APC model. The likelihood ratios between the APC model and the six other models were measured by deviances, Akaike Information Criterion, and degrees of freedom. This information was used to evaluate the goodness-of-fit of each model and to assess the significance of each factor. Statistical analyses were performed using SAS 9.4 (SAS Institute Inc., Cary, NC, USA). The *p*-values of <0.05 denoted statistical significance.

## 3. Results

Table 1 presents the incidence and prevalence of PsO and PsA from 2002 to 2016. A total of 112,865 PsO incident cases were identified (an average incidence rate of 34.21 per 100,000 persons). A decreasing pattern was observed from 43.33 per 100,000 persons in 2002 to 23.14 per 100,000 persons in 2016. In contrast, the incidence rates of PsA increased from 3.57 per 100,000 persons in 2002 and reached its peak to 6.76 per 100,000 persons in 2014. The prevalence of both PsO and PsA showed an increasing trend. The joinpoint regression analysis revealed significant changes in the linear trends. The incidence of PsO significantly decreased (an annual change of −2.8%, *p* < 0.001). In contrast, the PsO prevalence (an annual change of 3.4%, *p* < 0.001), PsA incidence (an annual change of 6.8%, *p* < 0.001), and PsA prevalence (an annual change of 11%, *p* < 0.001) all significantly increased.

The age-standardized incidence rates of PsO and PsA are shown in Figure 1. For PsO, this rate decreased from 2002 to 2016 (Figure 1a); for PsA, this rate continuously increased from 2002 to 2014, but decreased thereafter (Figure 1b). The age-standardized incidence of mild PsO showed a decreasing pattern. The corresponding incidence of severe PsO showed a bimodal distribution, with two peaks observed around 2002 and 2015. The patterns and values of mild PsO (Figure 1c) by calendar year were similar to those of PsO, suggesting the predominance of mild PsO. Similarly, the pattern of severe PsO (Figure 1d) was similar to that of PsA.

Figure 2 shows the age trends by the birth cohort, the period trends by age, and the birth cohort trends by the age of patients with PsO and PsA from 2002 to 2016. For the age trends (Figure 2a), the oldest birth cohort had the highest PsO incidence rate (17-fold higher than the youngest birth cohort). The long-term period trends of PsO incidence for the various age groups consistently decreased across the periods (Figure 2b). Similarly, the birth cohort trends for the incidence of PsO decreased consecutively since the 1937 birth cohort (Figure 2c). Of note, the trends for PsA were somewhat different from those of PsO. The age curves (Figure 2d) increased and stabilized after the 55–59 years age group. Increasing trends were observed across the periods (Figure 2e). The birth cohort trends for the incidence of PsA increased for the birth cohorts between 1937 and 1957 but decreased thereafter (Figure 2f).

Table 2 presents the goodness-of-fit for the APC model assessment. The likelihood-ratio test showed that the three-factor APC model had the lowest deviance (20.27 and 18.80), while the other six models (three one-factor and two-factor models) exhibited a significantly higher deviance. This indicated that the three-factor APC model was the best-fitted model to evaluate the trend in the incidence rates for PsO and PsA. In the two-factor models, the period–cohort model had the largest value of likelihood-ratio statistic, which suggest that aging has the strongest effect on incidence compared with period and cohort effects.

Figure 3 shows the best-fitted three-factor APC model estimates for the incidence of PsO and PsA. After adjusting for period and cohort effects, the net age effect on the incidence of PsO continuously increased with advancing age. The rate ratio (RR) for the incidence of PsO increased six-fold from the youngest age group (≤14 years) to the oldest age group (≥70 years). For the incidence of PsA, a steadily increasing trend was observed, which decreased after the 50–54 years age group. The RR of the 50–54 years age group was 7.7-fold higher than that noted in the ≤14 years age group. After adjusting for age and cohort effects, the net period effect exhibited a decreasing and increasing trend for PsO and PsA, respectively. During the observation period, the RR showed a decrease of approximately 25% for PsO, whereas it showed a 1.7-fold increase for PsA. After adjustments for age and period effect, the net cohort effect on PsO and PsA incidence continuously decreased from 1937. From the earliest to the most recent birth cohorts, the RR for the incidence of PsO and PsA showed a decrease of 43.4% and 72.1%, respectively. We also ran a three-factor APC model for mild and severe PsO, revealing similar trends for PsO and PsA (Figure 4).

## 4. Discussion

The present study revealed a significant decreasing and increasing trend in the incidence of PsO and PsA, respectively. The average incidence and prevalence of PsO and PsA were similar to those that have been reported in other regions of East Asia, but lower than those that have been reported in the USA, Europe, and Africa [21]. The difference may be explained by the lower frequency of the HLA-Cw6 gene in the Taiwanese population [4,22]. Previous data have demonstrated that the incidence of PsO either remains stable or decreases over time [6,23]. Moreover, the prevalence of PsA and PsO showed increasing trends that were compatible with those that have been reported in other countries [21]. Psoriatic disease is a chronic condition lacking a cure and the number of patients with this disease accumulates steadily over time. The increase in PsO/PsA prevalence may be related to a better awareness of the disease [24], reduction in mortality [11,25], or an actual increase in prevalence. The decreasing and increasing trend in PsO incidence and prevalence, respectively, suggests a decline in PsO-related mortality. Furthermore, longitudinal studies are warranted in order to examine this hypothesis.

This is the first study performing an APC analysis for the incidence of PsO and PsA over a period of 15 years using data from a comprehensive national-level database. Studies have demonstrated a bimodal distribution in PsO incidence/prevalence [21]. Consistent with previous studies in Taiwan, we did not find a similar distribution [5,26]. The net age effect showed that the incidence rate of PsO increased with advancing age (six-fold difference). The elderly are commonly afflicted with chronic disease (approximately 80% of older adults have one chronic illness and 50% have at least two chronic conditions) [27]. Diabetes, hypertension, hyperlipidemia, and obesity are significantly associated with an increased risk for PsO [28,29,30]. In Taiwan, over half of the elderly population are overweight (BMI of 24 to 27 kg/m^2^) or obese (BMI of ≥27 kg/m^2^) [31,32]. Thus, it is likely that the prevalence of chronic disease has contributed to the increased risk for developing new-onset PsO in the elderly. However, as the BMI data was not available in the database, we cannot evaluate the direct association between obesity and psoriasis.

The period effect could be influenced by a complex set of environmental, historical, social, and economic factors (e.g., war, famine, and economic crisis). The increasing awareness regarding PsO among physicians and the public and the frequent use of health services over the past decades should have led to more frequent identification of PsO. However, this study revealed a decreasing trend for the incidence of PsO after adjusting for the age and cohort effects. Most of the PsO cases were mild; hence, they may not seek medical care due to the easy accessibility of over-the-counter medicine in Taiwan. In the USA, the proportion of untreated mild PsO cases ranged 36.6–49.2% [33]. In addition, patients with psoriasis may visit the clinic less than once a year [34]. It remains unclear whether this “selection bias” indicates an underestimation or not. Despite that, we found a significant decreasing trend in the incidence of PsO, which may prove to be useful information worth noting. The net increasing trend in the incidence of PsA may be attributed to the enhanced knowledge regarding PsA by rheumatologists and dermatologists, the development of more sensitive screening tools [35], the publication of the Classification Criteria for Psoriatic Arthritis (CASPAR) in 2006 [36], and the reimbursement of effective biologics against PsA [37]. Moreover, severe PsO was associated with a higher risk of developing PsA [38]. James et al. [6] showed that approximately 50–60% of PsA diagnoses in Taiwan were related to PsO. In our study, the trend of severe PsO was mostly present for PsA. The mean duration of the transition from psoriasis to psoriatic arthritis was 3–5 years [6,39,40]. Thus, the increasing incidence of PsA was more associated with the increasing prevalence of PsO. This may explain the rising incidence rate of PsA, even though there was a decreasing trend for the incidence of PsO.

The cohort effect represents influences across groups of individuals that were born in the same year or generation. One’s birth cohort influences one’s health behaviors, as each cohort carries their own imprints of physical or social exposure and habits over the course of life [41,42,43]. The cohort effect revealed decreasing trends in the incidence of PsO and PsA from older to younger birth cohorts. This may be explained by higher health risks in the early birth cohorts. The younger birth cohorts have received good health education and have developed strong awareness compared with the early birth cohorts who may have adopted unhealthy lifestyles [44].

The strength of the APC analysis allowed us to examine the contribution of a particular period or birth cohort to the incidence of PsO and PsA. The exclusion of the effects of period and cohort on the incidence of disease according to age may influence the results [45]. This emphasizes the importance of the three-factor APC model in this analysis. Although the incident rates of PsO and PsA showed opposite trends, the incidence of both conditions remained high in the older population. The biggest social issue Taiwan confronts is that it is becoming an aging society, and changes in the structure of the population could affect the incidence of psoriatic disease. PsO is a multifactorial disease. Several lifestyle factors (e.g., smoking and alcohol consumption), stressful life events, obesity, cold weather, and diet have been associated with psoriatic disease [46]. Studies have shown that PsO negatively affects the societal productivity, work capacity, and socioeconomic status of patients [47,48], and increases their health care costs [49,50]. Furthermore, the gradient of rise in medical costs exceeds the increases in prevalence and incidence rates [51]. Therefore, it is imperative to conduct comprehensive projects on health-promoting behaviors (e.g., health education, regular physical activity, healthy diet, or obesity prevention) and effective screening strategies for the detection of PsO/PsA, particularly in middle-aged and older individuals. Delays in the diagnosis of PsA could result in significantly more radiographic damage and worse physical function [52]. Hence, collaboration between dermatologists and rheumatologists is recommended in order to manage PsA [53]. In addition, if pain and associated symptoms are unrelieved by non-steroidal anti-inflammatory drugs (NSAIDs) or disease-modifying antirheumatic drug (DMARD) therapy, complementary therapies, including granulocyte and monocyte adsorption apheresis [54] or acupuncture [55], have been suggested as alternative options.

There are some limitations to this study. Firstly, the incident and prevalent cases were assessed by diagnostic coding in an administrative claims database. In order to improve the diagnostic accuracy, we focused on psoriatic disease that was diagnosed by dermatologists and rheumatologists. Secondly, APC analysis is an ecological study. The result is an estimate, and we were unable to confirm a causal relationship. We attempted to elucidate the causality of these trends in the incidence of psoriatic disease based on the available data. Finally, health-seeking behavior may bias the interpretability of the data, as patients with mild PsO may not receive treatment. In addition, only the patients seeking care at a department of dermatology and rheumatology with three consecutive visits were included in the study. These factors could cause an underestimation of the incidence and prevalence of PsO/PsA.

## 5. Conclusions

This study showed a decreasing and an increasing trend in the incidence of PsO and PsA, respectively. The APC analysis revealed that the incidence of PsO and PsA is associated with older age and early birth cohorts. Therefore, the elderly individuals in Taiwan may be at a higher risk of developing new-onset PsO and PsA.

## Figures and Tables

**Figure 1 jcm-11-03744-f001:**
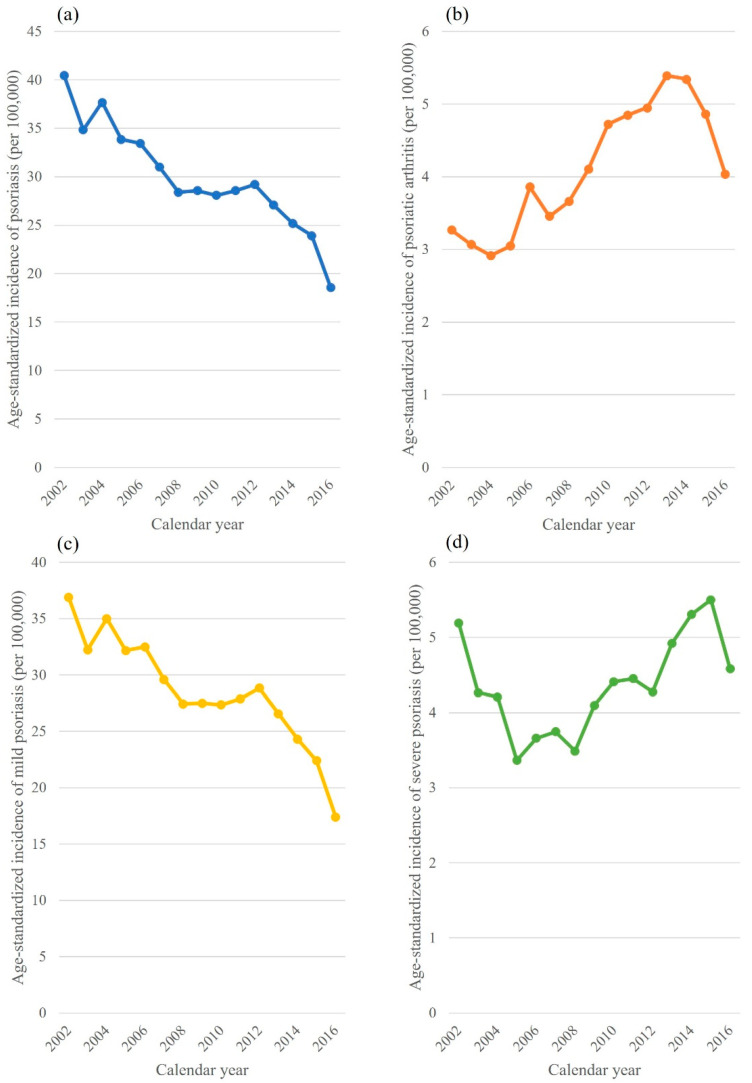
Age-standardized incidence rate of (**a**) psoriasis, (**b**) psoriatic arthritis, (**c**) mild psoriasis, and (**d**) severe psoriasis.

**Figure 2 jcm-11-03744-f002:**
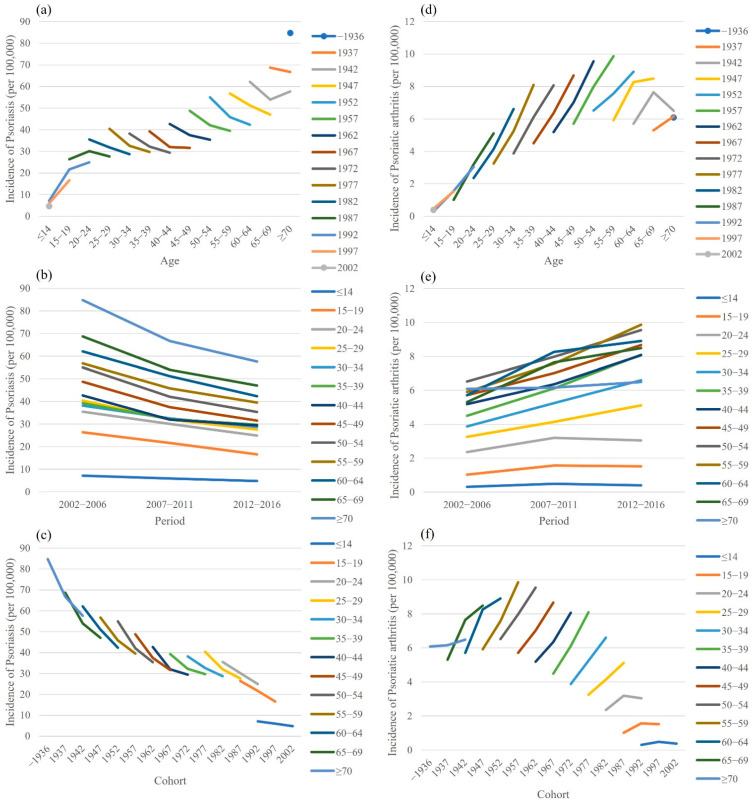
Incidence rates of psoriasis and psoriatic arthritis by age, period, and cohort. (**a**) age trends by cohort, (**b**) period trends by age, (**c**) cohort trends by age, (**d**) age trends by cohort, (**e**) period trends by age, and (**f**) cohort trends by age.

**Figure 3 jcm-11-03744-f003:**
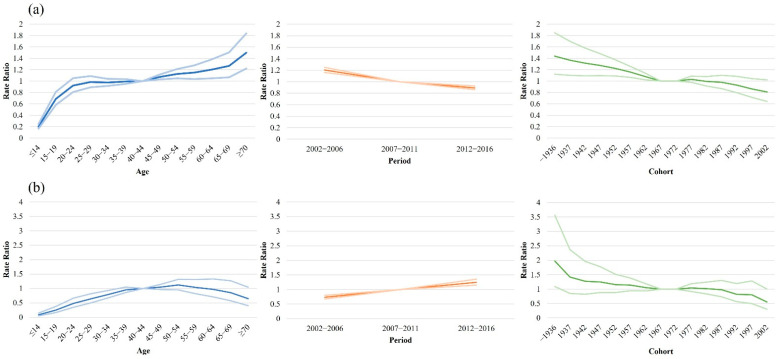
Three-factor age–period–cohort model analysis for incidence rate of (**a**) psoriasis and (**b**) psoriatic arthritis.

**Figure 4 jcm-11-03744-f004:**
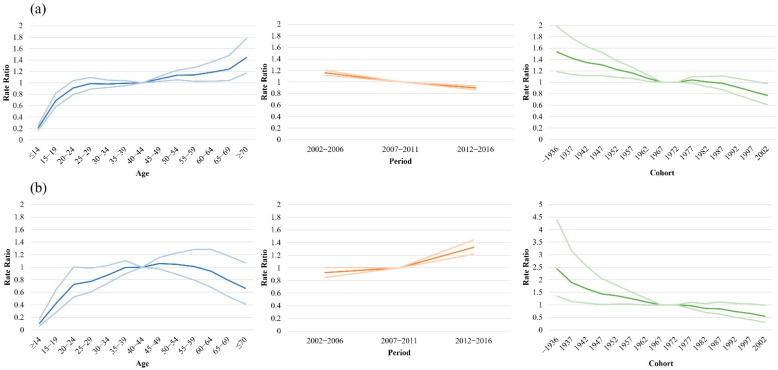
Three-factor age–period–cohort model analysis for incidence rate of (**a**) mild psoriasis and (**b**) severe psoriasis.

**Table 1 jcm-11-03744-t001:** Incidence and prevalence of psoriasis and psoriatic arthritis.

		Psoriasis				Psoriatic Arthritis			
Year	Population	Incident Case	Incidence Rate *	Prevalent Case	Prevalence Rate *	Incident Case	Incidence Rate *	Prevalent Case	Prevalence Rate *
2002	21,598,394	9358	43.33	29,049	134.50	771	3.57	1824	8.45
2003	21,452,868	8147	37.98	30,457	141.97	735	3.43	2064	9.62
2004	21,545,933	8880	41.21	33,702	156.42	713	3.31	2278	10.57
2005	21,369,558	7990	37.39	35,132	164.40	747	3.50	2479	11.60
2006	21,637,563	8080	37.34	36,759	169.89	956	4.42	2908	13.44
2007	21,816,951	7596	34.82	38,283	175.47	874	4.01	3109	14.25
2008	21,993,473	7070	32.15	38,786	176.35	940	4.27	3455	15.71
2009	22,116,920	7232	32.70	40,329	182.34	1067	4.82	3957	17.89
2010	22,176,505	7245	32.67	41,603	187.60	1235	5.57	4534	20.45
2011	22,263,419	7497	33.67	43,870	197.05	1318	5.92	5141	23.09
2012	22,362,328	7697	34.42	46,345	207.25	1350	6.04	5740	25.67
2013	22,477,022	7240	32.21	48,087	213.94	1503	6.69	6514	28.98
2014	22,564,915	6896	30.56	48,692	215.79	1525	6.76	7224	32.01
2015	22,633,061	6687	29.55	49,797	220.02	1383	6.11	7702	34.03
2016	22,692,606	5250	23.14	50,248	221.43	1184	5.22	7625	33.60
Average rate		34.21		184.29		4.91		19.96	
APC		−2.80		3.40		6.80		11.00	
95% CI		−3.7 to −2		3 to 3.8		5.4 to 8.2		10.2 to 11.8	
*p*-value		<0.001		<0.001		<0.001		<0.001	

* Incidence and prevalence rate: patients per 100,000 persons; APC, annual percentage change; CI, confidence interval.

**Table 2 jcm-11-03744-t002:** Goodness-of-fit of age–period–cohort model assessment for psoriasis and psoriatic arthritis incidence rates.

Models	df	Deviance	Likelihood-Ratio Statistic (df *)	*p*-Value
Psoriasis				
Age	26	2769.396	2749.1268 (15)	<0.0001
Period	36	32,449.15	32,428.8771 (25)	<0.0001
Cohort	24	3005.044	2984.7749 (13)	<0.0001
Age–Period	24	72.3037	52.0346 (13)	<0.0001
Age–Cohort	12	51.2918	31.0227 (1)	<0.0001
Period–Cohort	22	2824.326	2804.0565 (11)	<0.0001
Age–Period–Cohort	11	20.2691	reference	reference
Psoriatic arthritis				
Age	26	511.265	492.4695 (15)	<0.0001
Period	36	6474.931	6456.1351 (25)	<0.0001
Cohort	24	1553.344	1534.5484 (13)	<0.0001
Age–Period	24	68.0293	49.2338 (13)	0.0002
Age–Cohort	12	25.0532	6.2577 (1)	<0.0001
Period–Cohort	22	444.6162	425.8207 (11)	<0.0001
Age–Period–Cohort	11	18.7955	reference	reference

df: degree of freedom. Likelihood-ratio statistic: increase in deviance from the age–period–cohort model. df *: increase in df from the age–period–cohort model.

## Data Availability

Data was obtained from NHIRD and available with the permission of NHIRD.

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
