# Peer review of "Time Trends in Psoriasis and Psoriatic Arthritis Incidence from 2002 to 2016 in Taiwan: An Age–Period–Cohort Analysis"

_jcm, 2022, doi:10.3390/jcm11133744_

Round 1
Reviewer 1 Report
Thank you for submitting the manuscript. I have read your paper with great attention, which deals with an important and highly topical issue both in terms of research and in terms of assistance.The pathology you are facing and the possible complication are two entities that are difficult to classify and above all difficult to treat. You yourselves in the discussion mention the difficulty of treatment.Precisely to give greater prominence to the epidemiological data that you show, I suggest that you give greater emphasis in the discussion to the difficulty of treating psoriasis and psoriatic arthritis, which can also benefit from unconventional techniques.I suggest some references that can help you in the discussion:
doi: 10.1016/s0190-9622(03)02474-5.
doi: 10.1177/0964528420920281.
doi: 10.3109/09546630902817887.
I am convinced that this will improve the quality of your manuscript. I hope my comments are useful to you.
Kind Regards
Author Response
Dear Editors and Reviewers:
Thank you for your comments on our original research article entitled “Time Trends in Psoriasis and Psoriatic Arthritis Incidence from 2002 to 2016 in Taiwan: An Age-Period-Cohort Analysis.” We really appreciate the time, patience and efforts you made in scrutinizing our manuscript. We also appreciate the opportunity of minor revision. We tried our best to improve the manuscript and the manuscript has thoroughly been revised, and we herein respond to Reviewer’s comments individually, indicating how we have addressed each concern and describing the changes we have made (if applicable). Changes in the revised manuscript are detectable by track change. Below is the itemized, point-by-point response to the comments of the Reviewers.
Best regards,
Yu Tsung Chen,
On behalf of all authors
Reviewer 1
Thank you for submitting the manuscript. I have read your paper with great attention, which deals with an important and highly topical issue both in terms of research and in terms of assistance.The pathology you are facing and the possible complication are two entities that are difficult to classify and above all difficult to treat. You yourselves in the discussion mention the difficulty of treatment.Precisely to give greater prominence to the epidemiological data that you show, I suggest that you give greater emphasis in the discussion to the difficulty of treating psoriasis and psoriatic arthritis, which can also benefit from unconventional techniques.I suggest some references that can help you in the discussion:
doi: 10.1016/s0190-9622(03)02474-5.
doi: 10.1177/0964528420920281.
doi: 10.3109/09546630902817887.
I am convinced that this will improve the quality of your manuscript. I hope my comments are useful to you.
Kind Regards
Thank you for your comments. We have revised it.
Reviewer 2 Report
The authors present the age-period-cohort analysis. Enough data and statistical analysis are presented. This is a relevant topic and of great interest to both dermatologists and rheumatologists.
However, there are some points that should be improved in my opinion.
(1) Please improve Table 1 (Incidence and prevalence of psoriasis and psoriatic arthritis). Thick line is confusing.
(2) Fig. 3 & 4 should be color images.
(3) Please clarify why you chose this method of statistical analysis clearly.
(4) Although generally good, the manuscript benefit from careful proofreading by a native English speaker.
Thank you for the opportunity to read this manuscript.
Author Response
Reviewer 2
The authors present the age-period-cohort analysis. Enough data and statistical analysis are presented. This is a relevant topic and of great interest to both dermatologists and rheumatologists.
However, there are some points that should be improved in my opinion.
(1) Please improve Table 1 (Incidence and prevalence of psoriasis and psoriatic arthritis). Thick line is confusing.
(2) Fig. 3 & 4 should be color images.
(3) Please clarify why you chose this method of statistical analysis clearly.
(4) Although generally good, the manuscript benefit from careful proofreading by a native English speaker.
Thank you for the opportunity to read this manuscript.
Thank you for your comments. We have improved Table 1 and Figure 3&4. We also revised introduction to make it more clearly why we chose APC model. The manuscript has been edited by native English speaker.
Reviewer 3 Report
This is a well written and interesting paper on the incidence of PsO and PsA in Asian population. This is a very elegant investigation using a interesting methodology for analyzing the effect of age and cohort in the epidemiology of the disease. Some features need further discussion:
1. Under the introduction in the first paragraph, it is stated that psoriatic arthritis is characterized by oligoarthritis, etc, This is not absolutely true and varies with the different cohorts, so is better to change oligoarthritis for arthritis.
2. There are some epidemiologic studies in Taiwan that have not been mentioned: Wei JC-C, Shi L-H, Huang J-Y, Wu X-F, Wu R, Chiou J-Y. Epidemiology and medication pattern change of psoriatic diseases in Taiwan from 2000 to 2013: a nationwide, population-based cohort study. J Rheumatol 2018;45:385-92. doi:10.3899/jrheum.170516. There are also some systematic reviews on the epidemiology of Psoriasis and psoriatic arthritis that perhaps are worth reviewing.
3. The results are somehow surprising and, in some way, difficult to explain. The major risk factor for the development of psoriatic arthritis is in fact the presence of psoriasis. It is difficult to explain an increase in the incidence of psoriatic arthritis with a decrease in the incidence of psoriasis, even when the incidence of severe PsO increased. Severe PsO represents a small percentage of patients with PsO. In fact according to figure 1 the percentage of severe psoriasis among the incident cases is around 20%. Under discussion the increased prevalence is explained by increased awareness of the disease, however how would increased awareness increase the prevalence and not increase the incidence? A better explanation should be looked for, and perhaps should be searched in the methodology used.
4. Related to the previous point, it is not clear to me why the incident cases are not reflected in the prevalent cases. While you have around 8000 new cases of PsO per year the prevalence is increased only by around 2000 or 3000 patients. This means that new or all cases are moving away, or disappear from the system, could you please explain this?
5. Another major point to discuss is the low incidence and prevalence of PsA found in the study (more evident in the prevalence figures). The average prevalence around the world is 133100,000 person-years, and the lowest is 20/100,000 P/yrs. The average prevalence in your study was 19/100,000 p/years. This does not seem could be explained by ethnic characteristics, as a previous study in China found a prevalence of 200/100,000 p/yrs. Something similar, although less pronounced happened with the incidence of PsA. This, again, perhaps could be explained by methodological issues, and should be discussed.
Something similar, although much less evident occurred with the incidence and prevalence of PsO that are in the lower range of what has been reported.
6. The other surprising result was that the incidence of PsO increased with age.Most of the previous studies have shown that the incidence of PsO increases up to 39 years of age and then decreases, with a second peak between 50 or 60 years.. The explanation of the increase prevalence of comorbidities in this group is not accurate, as although the mentioned comorbidities are associated with PsO and PsA, the only one identified as a risk factor for the development of PsO and PsA is obesity. It is unlikely that the prevalence of obesity was increased in the older groups, so another explanation should be tried.
7. I do not agree with the statement that the misclassification bias should exert and equal effect on all calendar years. It is possible that with better treatments (including better use of topicals), patients do not attend medical consultations that often in the more recent years. As one of the inclusion criteria is to have at least three consecutive outpatient visits for psoriasis, many patients in more recent years might have been excluded.
8. In fact, and related to the previous point, have you analyzed how much the criteria of three consecutive outpatient visits improves the accuracy of the diagnosis in this database? The rationality of this inclusion criteria is not clear to me and perhaps could be explained.
Author Response
Reviewer 3
This is a well written and interesting paper on the incidence of PsO and PsA in Asian population. This is a very elegant investigation using a interesting methodology for analyzing the effect of age and cohort in the epidemiology of the disease. Some features need further discussion:
- Under the introduction in the first paragraph, it is stated that psoriatic arthritis is characterized by oligoarthritis, etc, This is not absolutely true and varies with the different cohorts, so is better to change oligoarthritis for arthritis.
Thank you for advice, we revised as your suggestion
- There are some epidemiologic studies in Taiwan that have not been mentioned: Wei JC-C, Shi L-H, Huang J-Y, Wu X-F, Wu R, Chiou J-Y. Epidemiology and medication pattern change of psoriatic diseases in Taiwan from 2000 to 2013: a nationwide, population-based cohort study. J Rheumatol 2018;45:385-92. doi:10.3899/jrheum.170516. There are also some systematic reviews on the epidemiology of Psoriasis and psoriatic arthritis that perhaps are worth reviewing.
Thank you for reminding, we have cited this paper in our article.
- The results are somehow surprising and, in some way, difficult to explain. The major risk factor for the development of psoriatic arthritis is in fact the presence of psoriasis. It is difficult to explain an increase in the incidence of psoriatic arthritis with a decrease in the incidence of psoriasis, even when the incidence of severe PsO increased. Severe PsO represents a small percentage of patients with PsO. In fact according to figure 1 the percentage of severe psoriasis among the incident cases is around 20%. Under discussion the increased prevalence is explained by increased awareness of the disease, however how would increased awareness increase the prevalence and not increase the incidence? A better explanation should be looked for, and perhaps should be searched in the methodology used.
Thanks for this constructive comment. Accordingly, mean duration to transition from psoriasis to psoriatic arthritis was 3-5 years. Thus, increasing incidence of PsA was more associated with increasing prevalence of PsO rather than incidence of PsO. This may explain the rising incidence rate of PsA, even though there was a decreasing trend for the incidence of PsO. As you said, increased awareness may increase the incidence, we discussed it in section of period and cohort effect. We also revised it.
- Related to the previous point, it is not clear to me why the incident cases are not reflected in the prevalent cases. While you have around 8000 new cases of PsO per year the prevalence is increased only by around 2000 or 3000 patients. This means that new or all cases are moving away, or disappear from the system, could you please explain this?
Thank you for accurate comment. As we mentioned in the article, most PsO cases were mild; after diagnosis as psoriasis, they may not continue visiting clinic due to the easy accessibility to over-the-counter medicine in Taiwan. And they won't be recored by the National Health Insurance Research Database. That's why there was a difference between incident and prevalent cases. Same condition was also founded in James' article.[1]
[1] Wei JC-C, Shi L-H, Huang J-Y, Wu X-F, Wu R, Chiou J-Y. Epidemiology and medication pattern change of psoriatic diseases in Taiwan from 2000 to 2013: a nationwide, population-based cohort study. J Rheumatol 2018;45:385-92.
- Another major point to discuss is the low incidence and prevalence of PsA found in the study (more evident in the prevalence figures). The average prevalence around the world is 133100,000 person-years, and the lowest is 20/100,000 P/yrs. The average prevalence in your study was 19/100,000 p/years. This does not seem could be explained by ethnic characteristics, as a previous study in China found a prevalence of 200/100,000 p/yrs. Something similar, although less pronounced happened with the incidence of PsA. This, again, perhaps could be explained by methodological issues, and should be discussed.
Something similar, although much less evident occurred with the incidence and prevalence of PsO that are in the lower range of what has been reported.
Thank you for your comments. In our results, the average incidence and prevalence of PsO and PsA were similar to those reported in other regions of east Asia, but lower than those reported in the USA, Europe, and Africa.[1] The difference may be explained by the lower frequency of the HLA-Cw6 gene in the Taiwanese population.[2,3] Our results were also similar to other studies in Taiwan population.[4,5] We have added this section to the Discussion.
[1] Parisi, R., et al., National, regional, and worldwide epidemiology of psoriasis: systematic analysis and modelling study. Bmj, 2020. 369: p. m1590.
[2] Raychaudhuri, S.P. and E.M. Farber, The prevalence of psoriasis in the world. J Eur Acad Dermatol Venereol, 2001. 15(1): p. 16-7.
[3] Chang, Y.T., et al., A study of candidate genes for psoriasis near HLA-C in Chinese patients with psoriasis. Br J Dermatol, 2003. 148(3): p. 418-23.
[4] Tsai TF, Wang TS, Hung ST et al. Epidemiology and comorbidities of psoriasis patients in a national database in Taiwan. J Dermatol Sci 2011; 63: 40-6.
[5] Chang YT, Chen TJ, Liu PC et al. Epidemiological study of psoriasis in the national health insurance database in Taiwan. Acta Derm Venereol 2009; 89: 262-6.
- The other surprising result was that the incidence of PsO increased with age.Most of the previous studies have shown that the incidence of PsO increases up to 39 years of age and then decreases, with a second peak between 50 or 60 years.. The explanation of the increase prevalence of comorbidities in this group is not accurate, as although the mentioned comorbidities are associated with PsO and PsA, the only one identified as a risk factor for the development of PsO and PsA is obesity. It is unlikely that the prevalence of obesity was increased in the older groups, so another explanation should be tried.
Thanks for this constructive comment. Our results were consistent with previous studies in Taiwan, we did not find a bimodal distribution.[1,2] A study published on Scientific showed that diabetes, hypertension, hyperlipidemia, and obesity were all found to be risk factors for incident psoriasis.[3] Obesity and hypertension, hyperlipidemia, and metabolic syndrome are significantly associated with an increased risk for PsO[4,5]. In Taiwan, over half of elderly are overweight (BMI: 24 to < 27 kg/m2) or obese (BMI ≥ 27 kg/m2)[6,7]. But as the data of BMI was not available in the database, we cannot evaluate the direct association between obesity and psoriasis. We have reworded the section.
[1] Tsai, T.F., et al., Epidemiology and comorbidities of psoriasis patients in a national database in Taiwan. J Dermatol Sci, 2011. 63(1): p. 40-6.
[2] Chang, Y.T., et al., Epidemiological study of psoriasis in the national health
[3] Kim ES, Han K, Kim MK et al. Impact of metabolic status on the incidence of psoriasis: a Korean nationwide cohort study. Sci Rep 2017; 7: 1989.
[4] Bodyweight variability and the risk of psoriasis: a nationwide population-based cohort study
[5] Kim, H.N.; Han, K.; Song, S.W.; Lee, J.H. Hypertension and risk of psoriasis incidence: An 11-year nationwide population-based cohort study. PLoS ONE 2018
[6] Association between obesity and education level among the elderly in Taipei, Taiwan between 2013 and 2015: a cross-sectional study.
[7] Directorate-General of Budget, Accounting and Statistics, Executive Yuan, R.O.C.(Taiwan). https://www.stat.gov.tw/public/Data/169152483HCL2D3O.pdf (chinese)(archived on 9 June 2021)
- I do not agree with the statement that the misclassification bias should exert and equal effect on all calendar years. It is possible that with better treatments (including better use of topicals), patients do not attend medical consultations that often in the more recent years. As one of the inclusion criteria is to have at least three consecutive outpatient visits for psoriasis, many patients in more recent years might have been excluded.
Thank you for your comment. We agreed with you and revised it.
- In fact, and related to the previous point, have you analyzed how much the criteria of three consecutive outpatient visits improves the accuracy of the diagnosis in this database? The rationality of this inclusion criteria is not clear to me and perhaps could be explained.
Thank you for this comment. This criteria has been validated[1,2] and used in many studies[3-5]. We agreed there may be a limitation, and we have mentioned it as a limitation.
[1] Ahlehoff O, Gislason GH, Jorgensen CH, Lindhardsen J, Charlot M, Olesen JB, et al. Psoriasis and risk of atrial fibrillation and ischaemic stroke: a Danish nationwide cohort study. Eur Heart J 2011;33: 2054–2064.
[2] Ahlehoff O, Gislason GH, Charlot M, Jørgensen CH, Lindhardsen J, Olesen JB, et al. Psoriasis is associated with clinically significant cardiovascular risk: a Danish nationwide cohort study. J Intern Med 2011; 270: 147–157.
[3] Tsai TF, Wang TS, Hung ST et al. Epidemiology and comorbidities of psoriasis patients in a national database in Taiwan. J Dermatol Sci 2011; 63: 40-6.
[4] Wei JCC, Shi LH, Huang JY, et al. Epidemiology and medication pattern change of psoriatic diseases in Taiwan from 2000 to 2013: a nationwide, population-based cohort study. J Rheumatol 2018;45:385-92.
[5] Chen YJ, Chang YT, Shen JL, et al. Association Between Systemic Antipsoriatic Drugs and Cardiovascular Risk in Patients With Psoriasis With or Without Psoriatic Arthritis. Arthritis Rheum 2012 Jun;64(6):1879-87.